

# Solar 27-day signatures in standard phase height measurements above central Europe

Christian von Savigny[1], Dieter H. W. Peters[2], and Günter Entzian[2]

[1]Institute of Physics, University of Greifswald, Felix-Hausdorff-Str. 6, 17489 Greifswald, Germany
[2]Leibniz-Institute of Atmospheric Physics e.V. at Rostock University (IAP), Schlossstraße 6, 18225 Kühlungsborn, Germany

*Correspondence to:* Christian von Savigny
(csavigny@physik.uni-greifswald.de)

**Abstract.** We report on the effect of solar variability at the 27-day and the 11-year time scale on standard phase height measurements carried out in central Europe. Standard phase height corresponds to the reflection height of radio waves in the ionosphere near 80 km altitude. Using the superposed epoch analysis (SEA) method, we extract statistically highly significant solar 27-day signatures in standard phase heights. The 27-day signatures are roughly anti-correlated to solar proxies, such as the F10.7 cm radio flux or the Lyman-$\alpha$ flux. The sensitivity of standard phase height change to solar forcing at the 27-day time scale is found to be in good agreement with the sensitivity for the 11-year solar cycle, suggesting similar underlying mechanisms. The amplitude of the 27-day signature in standard phase height is larger during solar minimum than during solar maximum, indicating that the signature is not only driven by photo-ionisation of NO. We identified statistical evidence for an influence of ultra-long planetary waves on the quasi 27-day signature of standard phase height in winters of solar minimum periods.

## 1 Introduction

The electromagnetic radiation emitted by the sun exhibits variability over a large range of different temporal scales. At time scales shorter than a century the most important solar variability cycles are the 11-year Schwabe-cycle (Schwabe, 1843) – being part of the 22-year Hale-cycle (Hale et al., 1919) – as well as the quasi 27-day solar cycle, which is caused by the sun's differential rotation (presumably first observed by Galileo Galilei or Christoph Scheiner in the first half of the 17th century). Note that the differential rotation of the sun does not lead to variations in solar proxies with a period of exactly 27 days. However, the term "27-day cycle" will be used in the following.



The 11-year solar cycle variation in total solar irradiance (TSI) amounts to only 0.1 % of its mean value of about 1361 W m$^{-2}$, but at UV wavelengths the relative variations can be significantly larger. Solar variability associated with the solar 27-day cycle is generally smaller than for the 11-year cycle, but for strong 27-day cycles the variation in solar proxies can exceed 50 % of the 11-year variation. Solar 27-day signatures have been identified in a number of parameters in the Earth's atmosphere, including mesospheric abundances of $H_2O$ (Robert et al., 2010; Thomas et al., 2015), OH (Shapiro et al., 2012; Fytterer et al., 2015), $O_3$ (Hood, 1986), O (Lednytskyy et al., 2017), noctilucent clouds (Robert et al., 2010; Thurairajah et al., 2017; Köhnke et al., 2018) and temperature (Hood, 1986; von Savigny et al., 2012; Thomas et al., 2015). In addition, effects of the sun's rotational cycle on planetary wave activity in the stratosphere were reported in several studies (e.g., Ebel et al., 1981). Indications for solar 27-day signatures in tropospheric clouds (Takahashi et al., 2010; Hong et al., 2011) and tropospheric temperature (Hood, 2016) were also presented.

Solar 27-day signatures are relatively easy to identify, if the analyzed time series are sufficiently long to allow suppressing other sources of variability that are typically significantly larger than the solar signature. However, attributing the solar signature to specific physical or chemical processes is often difficult.

In this study we employ indirect phase height – i.e. radio wave reflection height – measurements using a transmitter in central France and a receiver in northern Germany. We investigate the presence and characteristics of solar 27-day signatures in standard phase heights (SPH). The presence of an 11-year solar cycle signature in SPH measurements – with SPH minimum during solar maximum and vice versa – has been demonstrated in previous studies and the sensitivity of SPH to solar forcing at the 11-year scale has been quantified (e.g., Peters and Entzian, 2015). The reason for the anti-correlation between SPH and solar activity is thought to be the enhanced UV irradiance during solar maximum, leading to higher electron densities associated with enhanced photo-ionisation of NO, and consequently lower phase heights.

The paper is structured as follows. Section 2 provides a brief description of the SPH data set used in this study. In section 3 we give an overview of the superposed epoch analysis and significance testing approach employed here. Section 4 presents the main results on solar 27-day signatures in SPH data and in section 5 potential driving mechanisms and implications are discussed. Conclusions are provided in section 6.

## 2 Standard phase height data

The principle behind deriving SPH has been recently described in detail by Peters and Entzian (2015) and only the most important aspects are summarized here. Electromagnetic radiation at a frequency of 164 kHz (162 kHz since February 1986) is transmitted by a broadcasting station in Allouis in central France (47° N, 2° E) and received in Kühlungsborn in northern Germany (54° N, 12° E)





since February 1959. Assuming one-hop propagation, the detected signal corresponds to the phase relation between the ground wave and the sky wave reflected in the D-region of the ionosphere and allows calculating the indirect phase height at the reflection point. The distance between Allouis and Kühlungsborn is 1023 km. The reflection point of the signal is located over the Eifel-mountain (50°

N, 6° E, Germany). The SPH is defined as the reflection height at a fixed solar zenith distance of 78.4° (see Peters and Entzian (2015) for more detailed information). Panel a) of Figure 1 shows the derived daily SPH variation from February 1959 to February 2017 based on release R4 of standard phase height measurements derived under the application of a new diagnostic method and for an extended period (Peters et al., 2018). The 11-year solar cycle signature is clearly visible. Also

discernible is a negative long-term trend, which was determined by Peters and Entzian (2015) to be -114 m/decade for the period 1959 to 2009. This negative long-term trend is attributed to the shrinking of the middle atmosphere associated with its cooling (e.g., Peters and Entzian, 2015; Peters et al., 2017). Furthermore, a quasi-bidecadal oscillation was found at two different altitudes (OH* Meinel emissions at about 87 km and plasma scale height at about 80 km) in the mesopause region

in summer which are anti-correlated Kalicinsky et al. (2018).

## 3   Methodology

### 3.1   Power spectral analysis

A classical approach is used to identify solar-driven 27-day variations in detrended daily time series. In a first step, we apply a 41-day running mean and then calculate the anomaly as the deviation from

the running mean for SPH and proxies of solar activity, like Lyman-$\alpha$ and the F10.7 cm solar flux (see, e.g., Figure 2). In a second step, a power spectral analysis of the anomaly time series based on wavelet analysis (Torrence and Compo, 1998) was carried out in order to determine spectra of the solar Lyman-$\alpha$ (LYA) and the SPH time series (see Figure 3). The calculation of the power spectrum uses the first 16,384 days only, due to the restriction of the wavelet analysis to time series

whose number of data points corresponds to a power of 2. The solar Lyman-$\alpha$ spectrum shows a dominant band at a period of around 27 days, as expected, and an additional increase at about half that period (i.e., 13.5 days). Other proxies of solar variability have also been analysed. The spectra of the F10.7 cm solar flux (SFL) and sunspot number (SPN) look similar (not shown). The SPH spectrum includes a much broader spectrum. Strong signatures at periods between 55 days

and 27 days are identified, which are weakly damped by the 41-day running mean. A white noise component below the 27-day variability is also found. This shows that the SPH spectrum is not the result of solar variability only – via photo-ionisation by Lyman-$\alpha$ radiation – but includes other causes of variability, as the atmospheric processes discussed later.



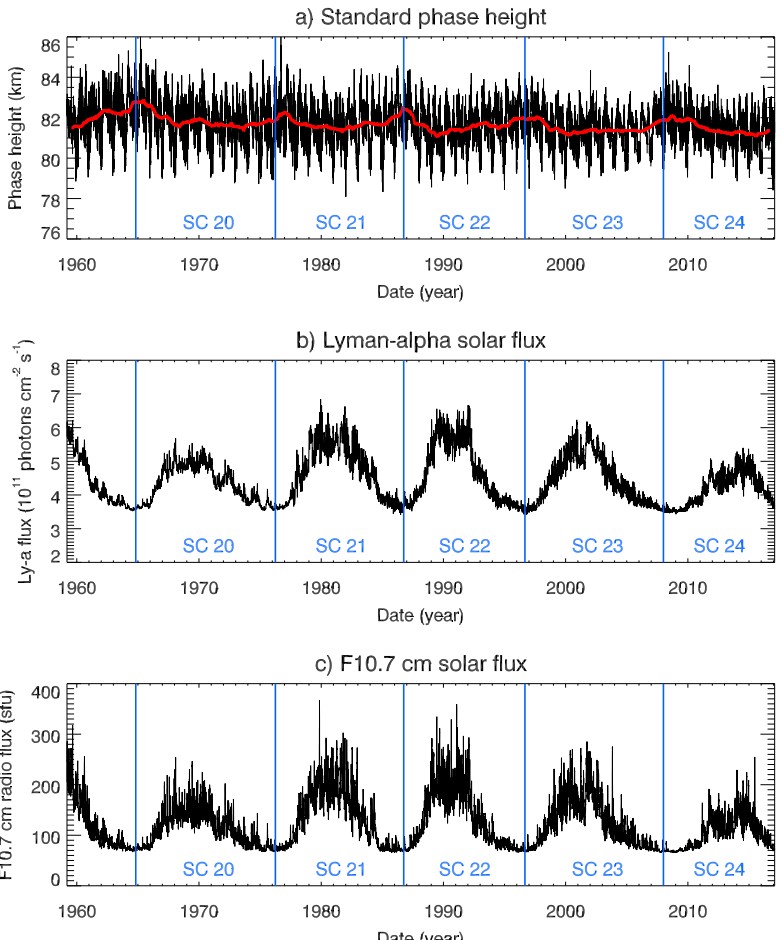

**Fig. 1.** Time series of standard phase height (top panel), solar Lyman-α flux (middle panel) and F10.7 cm solar flux (bottom panel) for the period from 02/1959 to 02/2017 ("SC" refers to "solar cycle"). The red line in the top panel corresponds to a 365-day running mean. The repeating pattern with a period of 1 year is the seasonal cycle in standard phase height data further discussed in Peters and Entzian (2015). An 11-year solar cycle signature is also discernible.

## 3.2 Superposed Epoch Analysis (SEA)

The analysis technique employed to extract solar-driven 27-day variations in standard phase height data is the superposed epoch analysis (further on referred to as SEA) technique (e.g., Howard, 1833; Chree, 1912), also known as composite analysis. The F10.7 cm solar radio flux or LYA is used as a solar proxy in the current study. Panels b) and c) of Figure 1 show LYA and the F10.7 cm solar flux



time series for the time period analyzed here, i.e., from February 1959 to February 2017.

In a first step we determined anomaly time series by removing a 41-day running mean from both the SPH and the F10.7 cm flux data. Using 41 days is arbitrary to a certain extent, but the results are only weakly dependent on the width of the smoothing window used, as will be discussed in more detail in section 4. Figure 2 shows the obtained anomaly time series for SPH (top panel) and the F10.7 cm solar flux (bottom panel). In order to quantify the variability of the two anomalies as a

function of time, we determined the standard deviation of the anomaly values in adjacent 100-day time bins. The red solid lines in the panels of Figure 2 display the time variation of these standard deviations. The standard deviation of the SPH anomaly is on the order of several hundred meters, which is significantly larger than solar 27-day signature extracted below using the SEA. Applying the same procedure to the LYA series, we found similar results, as expected (not shown).

Maxima in solar activity associated with the sun's differential rotation are identified automatically by searching for local maxima in the F10.7 cm flux time series smoothed with 5-day running mean filter. These local solar maxima are the centers of the analyzed epochs, each epoch covering 61 days, i.e., center date $\pm$ 30 days. Then the standard phase height anomalies for every epoch are written to the rows of a $N \times 61$ matrix, $N$ being the number of epochs analyzed. The main step of

the SEA consists of averaging the matrix column-wise, yielding the epoch-averaged standard phase height anomaly. Figure 4 shows the epoch-averaged F10.7 cm flux and the standard phase height anomalies for the entire data set from 1959 to 2017. The epoch-averaged F10.7 cm flux anomaly peaks at day 0 relative to local solar maximum, indicating that the epochs were selected correctly. The epoch-averaged standard phase height anomaly exhibits a periodic 27-day signature with an

amplitude of about 50 m and with a minimum occurring a few days before maximum solar activity. This finding is discussed below in section 4, where we also investigate the dependence of the SEA results on solar activity (applying different F10.7 cm flux thresholds) and on season.

### 3.2.1 Significance testing

Periodic signatures in the epoch-averaged anomalies may also be introduced by effects entirely un-

related to changes in solar forcing. A single major anomaly in the time series, e.g., related to a major stratospheric warming, will only cancel out in the analysis, if a sufficiently large number of epochs is available for analysis. Note that such an anomalous event may also lead to periodic variations in the epoch-averaged anomalies, if overlapping epochs are used, i.e., if the major anomaly occurs after local solar maximum in one epoch and before local solar maximum in the following epoch. In

other words, a repeating pattern in the epoch-averaged anomaly with a period of about 27 days is not necessarily an indication of the presence of a solar 27-day signature in the analyzed time series.

In order to test the significance of the obtained results, we applied the following Monte-Carlo test: Rather than choosing the epochs centered at local solar maxima, the epochs are chosen randomly, using the same number of epochs as for the actual analysis. The SEA with randomly selected epochs



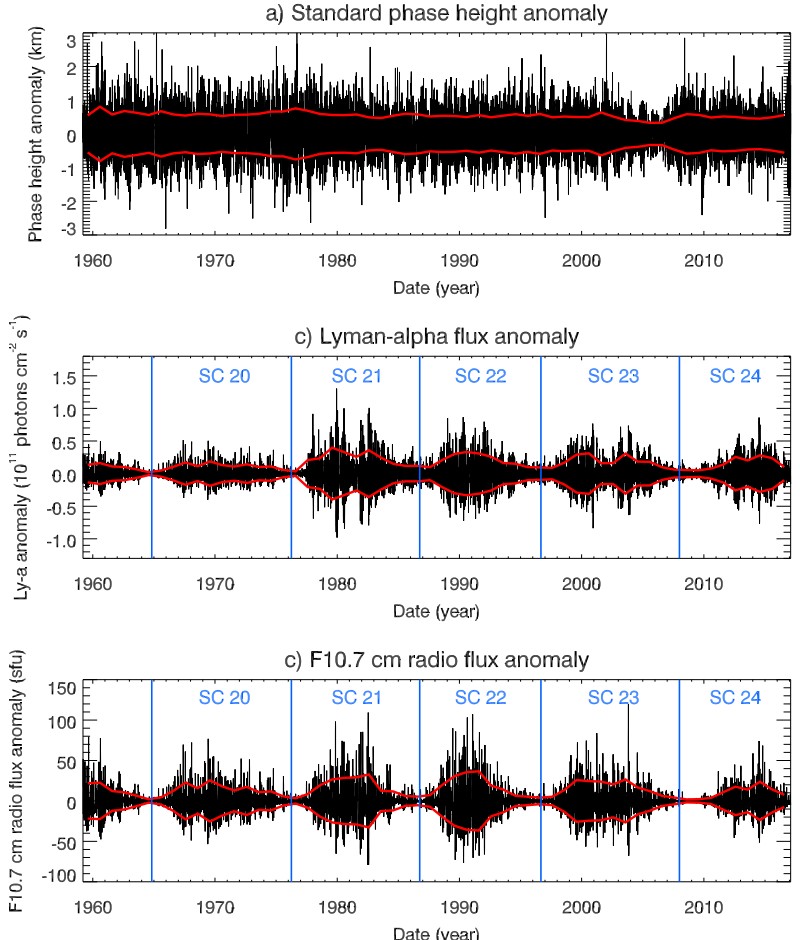

**Fig. 2.** Anomaly time series of standard phase height (top panel), solar Lyman-$\alpha$ flux (middle panel) and F10.7 cm solar flux (bottom panel), determined by removing a 41-day running mean from the time series shown in Figure 1. The red lines correspond to 1 standard deviation determined in adjacent 100-day time bins.

is performed 1000 times and a sinusoidal function is fitted to every single realization of the epoch-averaged standard phase height anomaly to determine its amplitude and phase. This is followed by checking in how many of the 1000 random cases the amplitude of the fitted sinusoidal function equals or exceeds the amplitude of the sinusoidal fit to the actual epoch-averaged standard phase height anomaly. Figure 5 shows as an example the amplitudes for the 1000 random realizations (in

black) and the amplitude of the actual SEA (in red). The amplitude of the signature in the actual SEA is not reached by any of the random realizations, indicating that the extracted 27-day signature




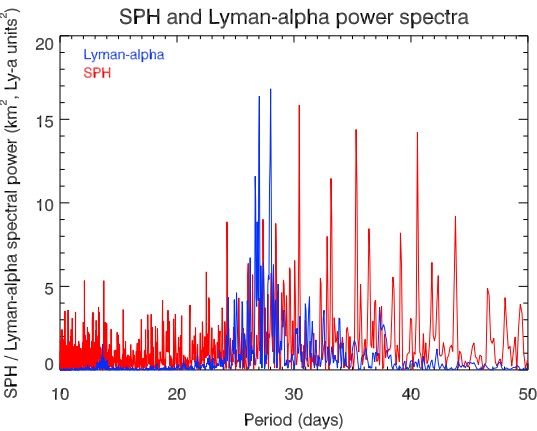

**Fig. 3.** Power spectra of the Lyman-alpha (LYA) and the standard phase height (SPH) time series starting in February 1959.

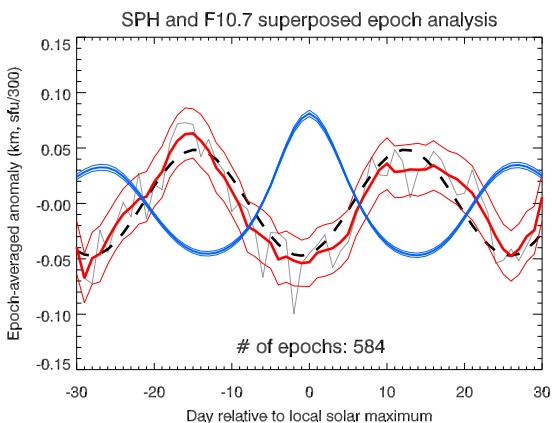

**Fig. 4.** Epoch-averaged F10.7 cm solar flux and standard phase height (SPH) anomalies for a total of 584 epochs. The thick blue line corresponds to the epoch-averaged F10.7 cm solar flux anomaly and the thin blue lines show the standard errors of the mean for each day relative to local solar maximum. The grey thin line corresponds to the unsmoothed epoch-averaged SPH anomaly, also shown smoothed by a 5-day running mean in red. The thin red lines represent the standard error of the mean of epoch-averaged anomalies about the daily mean value and plotted around the smoothed anomaly to improve clarity. The black dashed line is a sinusoidal fit to the unsmoothed epoch-averaged standard phase height anomaly, with an amplitude of about 50 m.

in standard phase height is very likely related to solar variability.




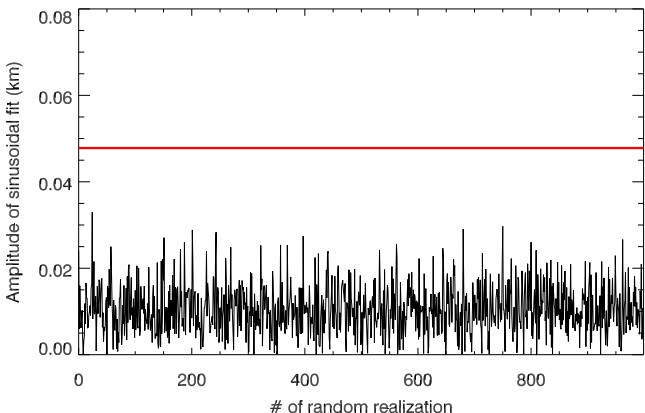

**Fig. 5.** Illustration of the Monte-Carlo significance test. The red line shows the amplitude of a sinusoidal fit to the extracted 27-day signature in SPH. The black line shows the fitted amplitudes to epoch-averaged SPH anomalies for 1000 randomly chosen epoch ensembles. See text for more detailed information.

## 4 Results

In section 4.1 we apply the band-pass filtering method based on wavelet analysis after Torrence and Compo (1998) in order to identify for the selected anomaly time series a comparable variability as the studied solar induced 27-day variation. The motivation comes from the result of the SEA (Figure 4) that already showed that a 27-day signature is present in the SPH time series which is strongly anti-correlated to the F10.7 cm solar flux or the Lyman-$\alpha$ flux. The used standard band-pass filter has a half width of about 10 % ($\sim$ 3 days) of the fundamental period of 27 days, i.e. a band-pass filter of 24 – 31 days is applied. These filtered time series are also used for a cross-correlation analysis. Furthermore, in section 4.2 we apply the superposed epoch analysis using F10.7 cm solar flux data in order to investigate the identified 27-day signature (Figure 4) in more detail. In section 4.3 we apply a regression analysis to the standard phase height time series as well ERA-Interim (Dee et al., 2011) and CMAM (McLandress et al., 2014) temperature and geopotential height time series to examine a possible link to atmospheric processes like planetary wave propagation and evolution.

### 4.1 Band-pass filtered time-series

The band-pass filtered (24 – 31 days) Lyman-$\alpha$ and SPH time series are shown in Figure 6 for three separated quasi-bi-decadal periods. In general, the LYA time series show a very high variability with different fluctuations during solar maximum and solar minimum. During solar maximum the LYA amplitudes are larger in solar cycles 21 and 22 than in solar cycle 20, for instance. Different seasonal SPH fluctuations are found for solar maximum in comparison to solar minimum. In general, the ratio between SPH amplitudes and LYA amplitudes is much larger than 1 during solar minimum with a





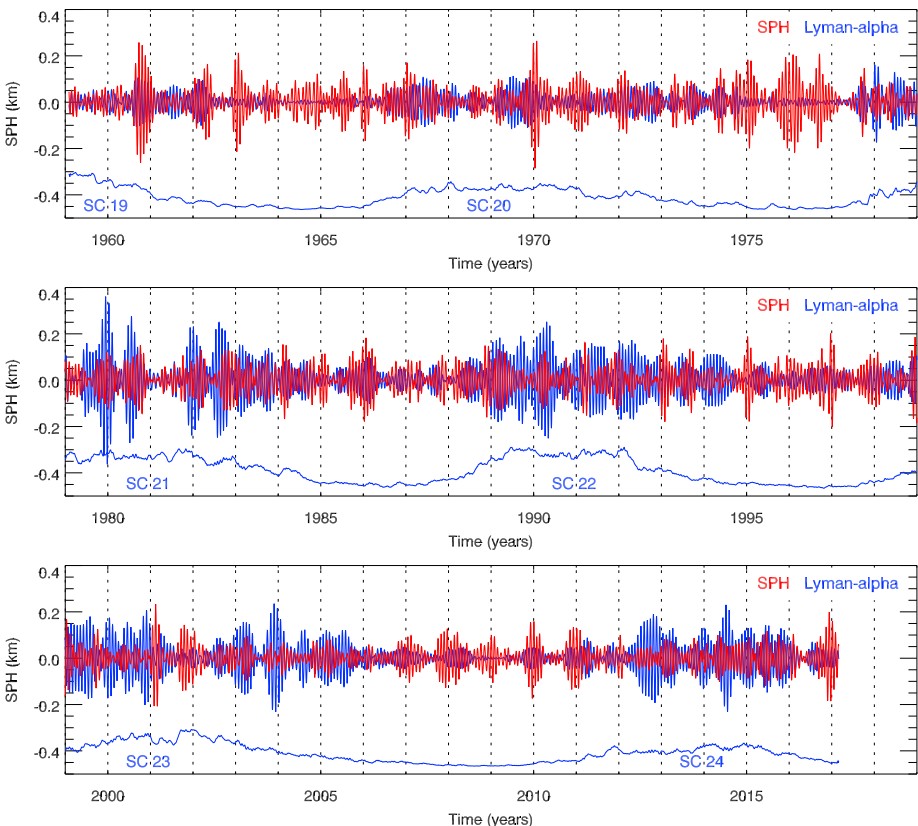

**Fig. 6.** Band-pass filtered (24 – 31 days) Lyman alpha (blue) and SPH (red) anomalies from 2/1959 to 2/2017 separated into three quasi-bi-decadal intervals. The solar cycles (SC 19 – SC 24) are also indicated by the smoothed 41-day running mean Lyman-$\alpha$ line (blue).

clear dominance for the winter months. Figure 7 shows as a typical example the winter 1985 – 1986 with an amplitude ratio exceeding 1 during solar minimum (note: with moderate LYA amplitudes

and larger SPH amplitudes). In summer the two band-pass-filtered time series are out-of-phase, as expected from photo-ionisation by Lyman-$\alpha$, but phase changes during winter time may be due to atmospheric processes.

In addition to Figure 7, the phase relationship is studied over the whole time series of 58 years. We examine the phasing between the SPH, and LYA, SFL, SPN anomaly series over all seasons.

The results of a cross correlation analysis (not shown) between those time series reveal a very weak anti-correlation between SPH and LYA, SFL, SPN, as expected. But the SPH shows a negative lag of 1 – 3 days for all three cross-correlations that means that on average the SPH minimum leads the maxima in solar activity proxies by a few days. This time lag is consistent with the time lag obtained





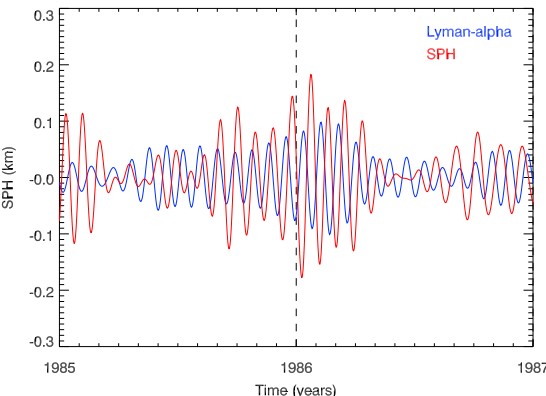

**Fig. 7.** As Figure 6 but from 1985 to 1987.

from applying the superposed epoch analysis (see section 4.2). Note that the cross-correlation was
run over all seasons and all 58 years. This result supports the hypothesis that atmospheric processes
determine the mean cross-correlation and finally the variability of SPH. Especially in winter during
solar minimum it seems that atmospheric processes are dominant.

### 4.2   Superposed epoch analysis

The SEA result displayed in Figure 4 already demonstrated that a 27-day signature is present in the
SPH time series. The Monte-Carlo significance test described above showed that the fitted amplitude
to the epoch-averaged SPH anomalies did not reach the actual amplitude for any of the 1000 random
ensembles, indicating that the 27-day signature in SPH in Figure 4 is very likely caused by the solar
27-day cycle. The 27-day signature in SPH has an amplitude of about 50 m and is thus significantly
smaller than the overall SPH variability (see bottom panel of Figure 2).

The SEA was so far applied to the entire time series covering the period from 02/1959 to 02/2017
and for a window width of 41 days when determining the anomaly time series. In the following sub-
sections we investigate, how the SEA results depend on solar activity (applying different thresholds
for the F10.7 cm flux), on season, and on the width of the window. As will be seen, the sensitivity
values are dependent on all of these assumptions.

**4.2.1   Sensitivity of standard phase height to the 27-day and 11-year solar cycles**

The sensitivity parameter (or simply sensitivity) that quantifies the SPH dependence on solar activity
is easily determined using the epoch-averaged F10.7 flux and SPH anomalies displayed in Figure 4.
The anomalies are plotted against each other in a scatter plot and the sensitivity parameter is given
by the slope of a linear regression line. Before the linear regression is performed, we determine the





phase lag between solar maximum and SPH minimum using time-lagged cross correlation. The SPH
anomaly is then shifted by the corresponding time lag (4 days for the results displayed in Figure 4),
followed by the linear regression. For a smoothing window width of 41 days and considering all
available epochs a sensitivity of -0.365 $\pm$ 0.043 km $(100\,\mathrm{sfu})^{-1}$ is obtained.

We also determined the sensitivity of the SPH to the 11-year solar cycle. This is done by defining
a regular F10.7 cm flux grid with a step size of 10 sfu, followed by averaging all daily F10.7 cm solar
flux values – and the corresponding SPH values – for each 10 sfu bin. The resulting bin-averaged
solar flux and SPH values are then plotted in a scatter plot and the sensitivity is given by the slope
of a line fitted by linear regression. The obtained value of the standard phase height sensitivity to
solar forcing at the 11-year time scale is -0.436 ($\pm$ 0.049) km $(100\,\mathrm{sfu})^{-1}$. This value agrees within
combined uncertainties with the standard phase height sensitivity for the 27-day solar cycle of -
0.365 ($\pm$ 0.043) km $(100\,\mathrm{sfu})^{-1}$, which suggests similar driving mechanisms. This aspect will be
discussed further in section 5. We also note that the 11-year SPH sensitivity derived here is in good
agreement with the value based on the results by Peters and Entzian (2015) of -0.387 km $(100\,\mathrm{sfu})^{-1}$.
Peters and Entzian (2015) used the Lyman-$\alpha$ flux as solar proxy, so that a conversion to the F10.7 cm
flux was required to convert their sensitivity value to units of km $(100\,\mathrm{sfu})^{-1}$. This was done using
a linear fit to the Lyman-$\alpha$ flux as a function of F10.7 cm radio flux (see Figure 8) for all available
data between 02/1959 and 02/2017.

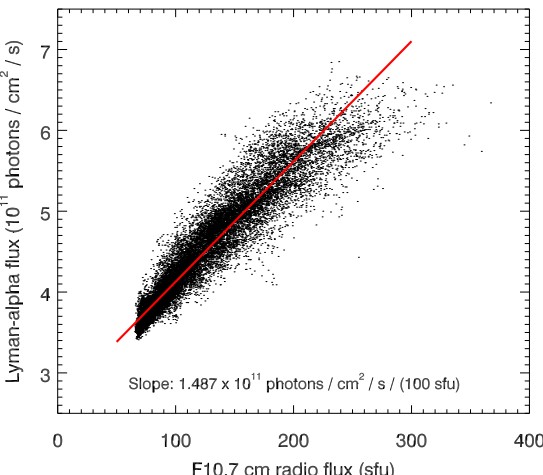

**Fig. 8.** Scatter plot of daily values of Lyman-$\alpha$ flux and F10.7 cm radio flux for the period from 02/1959 to
02/2017. The red line corresponds to a linear regression to the data points.



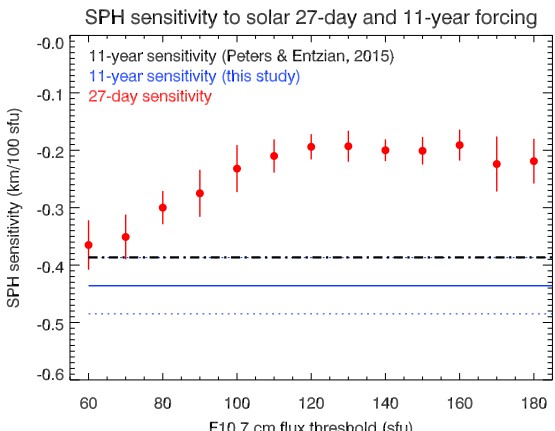

**Fig. 9.** SPH sensitivity to solar forcing for the 27-day and the 11-year solar cycle. The red circles show the 27-day sensitivity for different solar activity thresholds as described in the text. The blue line corresponds to the 11-year sensitivity determined in this study and the dotted line show the uncertainties. The black dash-dotted line displays the 11-year sensitivity determined by Peters and Entzian (2015).

### 4.2.2 Dependence of results on solar activity

Different tests were performed to study the dependence of the 27-day sensitivity of SPH on solar activity. First, we apply different solar activity thresholds (from 60 sfu up to 200 sfu in steps of 10 sfu) and only consider epochs for which the solar activity exceeds the assumed threshold on all days. The results of this test are listed in Table 1 and the dependence of the derived 27-day SPH sensitivity on solar activity is shown in Figure 9. Table 1 lists the number of epochs available for the different solar activity thresholds, the temporal shift applied before performing the linear fit, the amplitude of the fitted sine, as well as the results of the significance tests and the 27-day sensitivity value. The number of epochs decreases with increasing solar activity threshold, as expected. For the lowest three solar activity thresholds, the significance test did not yield a single random ensemble with amplitudes exceeding the amplitude obtained in the actual analysis. The fraction generally increases with increasing solar activity threshold and reaches about 35 % for a F10.7 cm flux threshold of 200 sfu. The shift or time lag varies somewhat between -1 and -4 and the negative sign implies that the minimum in standard phase height precedes the maximum in solar activity. The reasons for this behavior are currently not well understood and will be discussed in section 5.

   Figure 9 illustrates that the SPH sensitivity to solar forcing at the 27-day time scale depends on the solar activity threshold, but no simple or monotonous dependence is obvious. The Figure also displays the SPH sensitivity to solar forcing for the 11-year solar cycle. The blue line shows the value determined in this study (including uncertainties shown as blue dotted lines), and the black



| F10.7 flux threshold (sfu) | # of epochs | Shift (days) | Amplitude* (m) | Fraction† (%) | 27-day sensitivity (km (100 sfu)$^{-1}$) |
|---|---|---|---|---|---|
| 60 | 584 | -4 | 47.8 | < 0.1 | -0.365 ± 0.043 |
| 70 | 573 | -4 | 47.0 | < 0.1 | -0.351 ± 0.039 |
| 80 | 511 | -3 | 42.2 | < 0.1 | -0.300 ± 0.029 |
| 90 | 448 | -4 | 37.2 | 0.1 | -0.275 ± 0.041 |
| 100 | 398 | -4 | 33.4 | 0.3 | -0.232 ± 0.041 |
| 110 | 370 | -4 | 34.1 | 0.7 | -0.210 ± 0.029 |
| 120 | 330 | -3 | 34.2 | 1.4 | -0.194 ± 0.022 |
| 130 | 303 | -2 | 31.5 | 2.8 | -0.193 ± 0.027 |
| 140 | 267 | -3 | 40.5 | 0.4 | -0.200 ± 0.019 |
| 150 | 241 | -3 | 41.3 | 1.3 | -0.201 ± 0.024 |
| 160 | 221 | -3 | 36.3 | 3.6 | -0.191 ± 0.027 |
| 170 | 197 | -4 | 42.2 | 1.6 | -0.224 ± 0.048 |
| 180 | 171 | -4 | 41.0 | 3.4 | -0.219 ± 0.039 |
| 190 | 146 | -4 | 34.2 | 14.6 | -0.227 ± 0.059 |
| 200 | 126 | -5 | 27.6 | 34.7 | -0.598 ± 0.640 |

**Table 1.** Overview of the results for different solar activity thresholds (*Amplitude of fitted sinusoidal function; †Fraction of random realizations with amplitudes larger than actual data).

dash-dotted line corresponds to the value determined in the study by Peters and Entzian (2015), based on the same SPH data set.

Next, we tested how the results differ between periods of low and enhanced solar activity. This was done by selecting epochs for which the F10.7 cm flux was either lower or greater than 130 sfu. The epoch-averaged SPH anomaly for F10.7 > 130 sfu is shown in the upper panel of Figure 10 and the one for F10.7 < 130 sfu in the bottom panel of this Figure. Surprisingly, the amplitude of the extracted solar 27-day signature in SPH is larger for low solar activity than for higher solar activity. Because the absolute amplitude of the 27-day F10.7 cm flux variations for low solar activity is significantly smaller than during solar maximum, the SPH sensitivity to solar forcing at the 27-day scale and for low solar activity is with a value of -1.54 ± 0.38 km (100 sfu)$^{-1}$ also significantly larger than the value reported above.

### 4.2.3 Dependence of results on season

In addition, we investigated whether the solar 27-day signature in standard phase height depends on the season. For this purpose we consider "winter" to include the months October, November, December, January and February. "Summer" includes May, June, July, August and September. We use more than 3 months for each season in order to increase the number of epochs available for analysis. The smoothing window width is again 41, as above. The analysis results are listed in Table





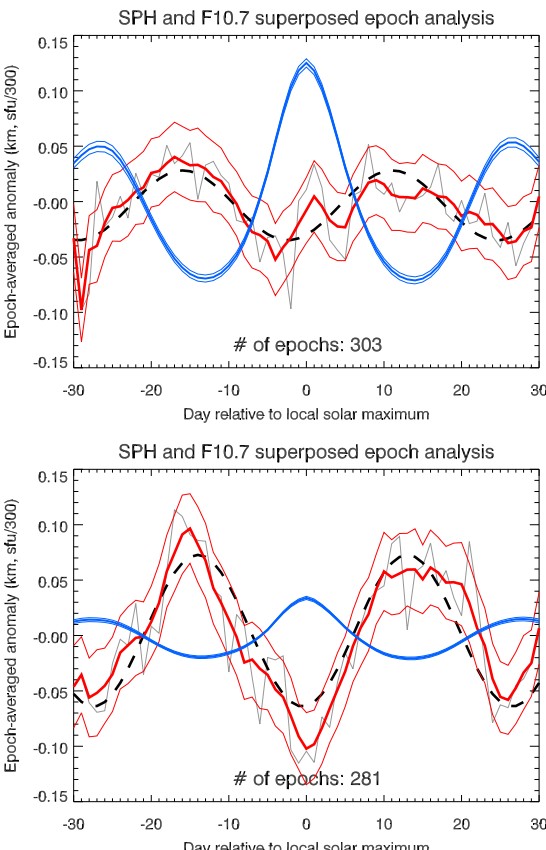

**Fig. 10.** Top panel: similar to Figure 4, but for epochs with solar activity exceeding 130 sfu. Bottom panel: similar to Figure 4, but for epochs with solar activity lower than 130 sfu.

2. The number of epochs used for the summer (243) and winter (244) seasons is almost identical and

250 the phase shift only differs by one day. However, the obtained amplitude is about a factor of 2 larger for the winter season than for summer. The SPH 27-day sensitivity for summer (-0.454 ± 0.077) agrees within uncertainties with the all-year value (-0.365 ± 0.043), but for winter, the value is with -0.488 ± 0.052 slightly larger. Potential reasons for this behavior are discussed below in section 5.

#### 4.2.4 Dependence of results on window width

250 Next, we tested the effect of different smoothing windows – used to determine anomaly time series – on the results. The window width (w) was increased from 30 days to 80 days in steps of 5 days. The obtained SPH sensitivities to solar forcing at the 27-day scale changed from -0.343 ($\pm$ 0.029) km (100 sfu)$^{-1}$ (w = 30 days) to -0.405 ($\pm$ 0.047) km (100 sfu)$^{-1}$ (w = 80 days). The dependence of the





| Season | # of epochs | Shift (days) | Amplitude* (m) | Fraction† (%) | 27-day sensitivity (km (100 sfu)$^{-1}$) |
|---|---|---|---|---|---|
| All year | 584 | -4 | 47.8 | < 0.1 | -0.365 ± 0.043 |
| Summer | 243 | -3 | 27.2 | 11.2 | -0.454 ± 0.077 |
| Winter | 244 | -3 | 54.9 | < 0.1 | -0.488 ± 0.052 |

**Table 2.** Overview of the results for different seasons (*Amplitude of fitted sinusoidal function; †Fraction of random realizations with amplitudes larger than actual data).

sensitivity on window width is not truly monotonous, but larger window widths have a tendency to
be associated with larger absolute sensitivity values. Changing the window width by 10 days, leads
to average changes in sensitivity of about 0.01 km (100 sfu)$^{-1}$, corresponding to a relative change
of about 3 %. We can therefore conclude, that the obtained sensitivities are only weakly dependent
on the smoothing window width.

### 4.3 Comparison of standard phase heights with ERA-I and CMAM

In subsection 4.3.1, we compare the variability of three data sets: the SPH time series, measured
over the Eifel mountain (50° N, 6° E; Western Germany) at about 82 km altitude (details are de-
scribed in section 2), temperature profiles averaged over the Eifel mountain region (40° – 58° N, 0°
– 12° E) from ERA-Interim data (Dee et al., 2011) and from the Extended Canadian Middle Atmo-
sphere Model (CMAM-Ext, CMAM30 results, McLandress et al. (2014)). Model data were down-
loaded from the following web-page: http://climate-modelling.canada.ca/climatemodeldata/cmam/-
cmam30/era_interim_adjustment/index.shtml Note that the CMAM-Ext model is nudged up to 1 hPa
with ERA-Interim data, i.e., CMAM-Ext and ERA-Interim show a similar temporal evolution in the
troposphere and stratosphere.

In addition, in subsection 4.3.2 we apply a regression analysis between the SPH time series and the
3-dimensional geopotential height field (GH) taken from CMAM, in order to examine the possible
link between SPH evolution (band-pass filtered) and the hemispheric variability of the planetary
wave field on a daily basis.

### 4.3.1 Comparison of time-series over Eifel-mountain

The ERA-Interim (red) and CMAM (blue) temperature evolutions at about 1 hPa over the Eifel
mountain region are in good agreement, as shown in the upper panel of Figure 11. This is expected,
because of the nudging procedure used in CMAM. This is demonstrated as an example for the decade
from 1979 to 1989. A stratopause warming is found in each summer season and a highly disturbed
winter evolution mainly due to the action of planetary waves. Results of a band-pass filter analysis
are shown in the lower panel of Figure 11 indicating large amplitudes of about 1 – 5 K for a 24 –



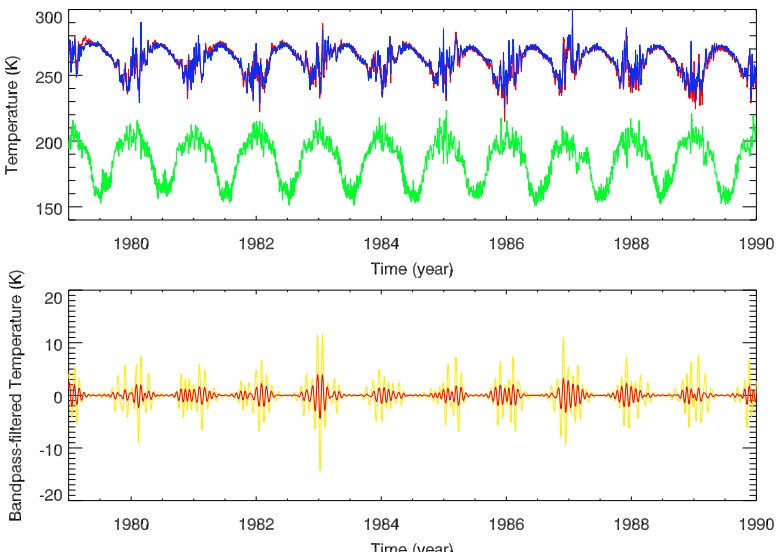

**Fig. 11.** Top part: Temporal evolution of ERA-Interim (red) and CMAM (blue) temperature at about 1 hPa (about 48 km) and CMAM temperature (green) at 0.01 hPa (about 80 km) averaged over the Eifel mountain region from January 1, 1979 to December 31, 1989. Bottom part: Band-pass filtered CMAM temperatures for a 24 − 31 day filter (red) and a 20 − 40 day filter (yellow).

31 day filter during winter and up to 10 K in the broader 20 − 40 day-filter band. Both filter bands are in phase but show different amplitudes for different boreal winters. That means that especially in winter a high variability in the stratopause temperature occurs with a dominant signal in 24 − 31 day filter band, which refers to the enhanced ultra-long planetary wave activity.

In the middle panel of Figure 11 the CMAM temperature evolution is shown at 0.01 hPa. The low-
est temperatures are found in summer – as known, e.g., from OH* rotational temperature measurements at Wuppertal (51° N, 7° E; Kalicinsky et al. (2016)) – and an anti-correlation to stratopause temperature. In each winter we found also a strong anti-correlation in the temperature variability between both layers induced by planetary wave activity which appears also in other meteorological fields due to the quasi-geostrophic balance. This ultra-long wave activity extends into the meso-
sphere as known from to the vertical propagation of ultra-long planetary waves (Charney and Drazin, 1961) in an eastward directed background flow including vacillation cycle behaviour as shown by Holton and Mass (1976). The hemispheric structure of ultra-long wave oscillations in the 24 − 31 day-filter band is investigated in the next subsection.





### 4.3.2 Regression of standard phase heights and CMAM geopotential heights

Following classical textbooks (e.g., Taubenheim, 1969) the regression between two times series is defined by the correlation between both multiplied by standard deviation of the first series and divided by the standard deviation of the second series. A time lag (lead) is introduced by a negative (positive) shift of days for the first time series and repeated calculation, respectively. As the second time series we choose the $24 – 31$ day band-pass filtered SPH and as the first time series we

use the non-filtered GH anomaly available at 64 longitudes (covering the range from $0°$ to $360°$ in steps of $5.625°$) and at 17 latitudes (between $0°$ N and $85.76°$ N in steps of about $5°$) and at each model pressure layer from the surface up to 0.001 hPa (62 layers). Selected results of the regression coefficient calculation are shown in Figure 12 for lags of 12 and zero days in order to examine the periodic behavior. The results are presented for 0.01 hPa (about 80 km, upper mesosphere, about the

layer of SPH measurements, panels a) and b)) and for 1 hPa (about 48 km, panels c) and d)) near the stratopause. All plots show extended regions of positive and of negative regression coefficients indicating large scale structures of similar regression as expected from the action of ultra-long planetary waves (about wave 1 to 3). In the upper mesosphere and for a lag of 12 days (Panel a) of Figure 12) we found a positive geopotential height (GH) anomaly of 300 m for 1 km SPH change over central

Europe. For a zero day lag (Panel b) of Figure 12) – i.e., about half a 27-day solar period later – a negative regression was found in a similar order. That means that over central Europe a GH change 12 days before is positively correlated with band-pass filtered SPH variability. 12 Days later there is a negative correlation. The hemispheric patterns are comparable indicating an ultra-long wave structure.

At 1 hPa and for a lag of 12 days (panel c) of Figure 12) we found a positive geopotential height (GH) anomaly of 500 m for 1 km SPH change over the central North Atlantic. For zero day lag (panel d) of Figure 12) – i.e., half a 27-day solar cycle later – a negative regression was found in a similar order. That means that over the central North Atlantic a GH change 12 days before is positively correlated with band-pass filtered SPH variability at 80 km altitude. 12 days later the

correlation is negative. In the stratopause region (Figure 12, panels c) and d)) the planetary wave regression patterns are more intense showing statistically significant correlations.

In the upper mesosphere, the cause for the negative regression pattern between GH anomaly and the $24 – 31$ day-band-pass filtered SPH time series over central Europe in about 80 km altitude for lag zero may be explained by an increase of NO density caused by southward transport of NO by

ultra-long waves in an observed mean positive latitudinal NO gradient, in a region between high and low pressure respectively.

It follows an increase of the free electron partial pressure due to photo-ionisation as discussed by von Cossart and Entzian (1976). If for a positive (negative) GH anomaly the electron density is higher (lower) including a lower (higher) layer of constant electrons (SPH) it follows a negative

regression pattern.



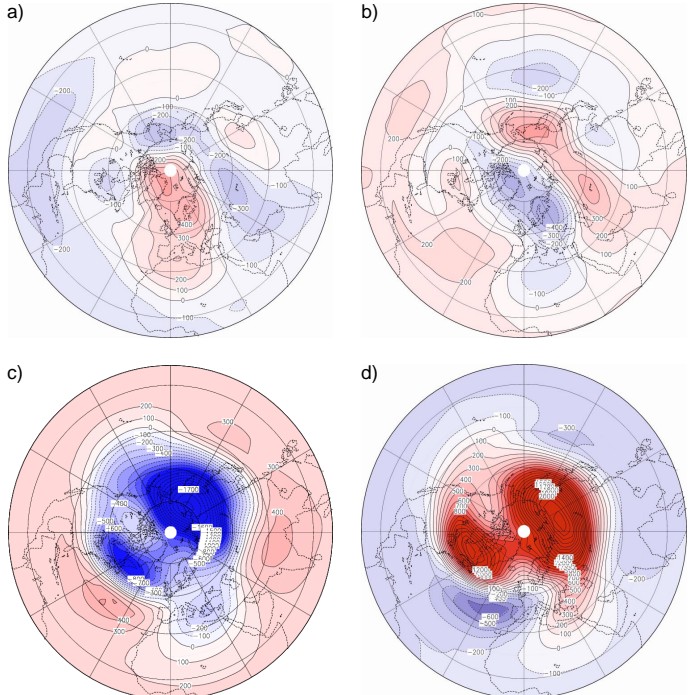

**Fig. 12.** Regression (in m/km) between the SPH (24 – 31 day-filtered) time series and the CMAM geopotential height time series at 0.01 hPa (a, b) and at 1 hPa (c, d) for time lags of 12 days (a, c) and 0 days (b, d) for the period 10/1985 to 4/1986.

If the air pressure is higher (lower) for positive (negative) GH anomaly, then the electron density is higher (lower) that means that the layer of constant electrons density (SPH) is shifted downward (upward). This hypothesis has to be examined in an atmospheric general circulation model including a chemistry and ion model, which is beyond the diagnostic study presented here. The positive

correlation pattern for a lag of 12 days (Figure 12, panel a) follows from the quasi-periodic 27-day oscillation behavior of the ultra-long wave structure. Furthermore, the negative regression for lag of 12 days and the positive regression for lag zero over eastern Europe reveal the cyclic evolution of the ultra-long planetary waves. In the stratopause layer the regression pattern is positively (negatively) correlated to the 24 – 31 day-band-pass filtered SPH time series over the polar region for lag zero

(12 days) – see panels c) and d) of Figure 12 – indicating a polar vortex weakening (strengthening). The vortex weakening is linked with an intrusion of subtropical air into the polar region over the North Atlantic, as known from some major stratospheric warming events in wintertime (e.g., Peters et al., 2014). A dominant wave 1 pattern occurs with a strong wave 2. In general the results reveal an atmospheric influence especially of ultra-long planetary waves on the 24 – 31 day-band-pass filtered



SPH time series during wintertime and solar minimum.

## 5   Discussion

In this study we investigated variability in SPH at temporal scales close to the solar 27-day cycle.
Different analysis techniques – i.e. cross-correlation analysis and superposed epoch analysis – were
applied to extract a potential solar driven 27-day signature in SPH data covering almost six solar
cycles.

The SEA, when applied to the entire SPH data set, yields evidence for a clear periodic 27-day
signature with an amplitude of about 50 m, which is very likely caused by the solar 27-day cycle,
as demonstrated by a Monte-Carlo significance analysis. An independent piece of evidence indi-
cating that the identified 27-day signature in SPH is caused by solar forcing, is the finding that the
determined SPH sensitivity to solar variability at the 27-day scale is good agreement with the sen-
sitivity for the 11-year solar cycle. SPH is more or less anti-correlated to solar forcing, which is
consistent with the simple picture that enhanced photo-ionisation of NO leads to an increase in free
electron density and subsequently to a decrease in SPH. However, several of our findings cannot be
reconciled with a purely photochemical mechanism.

First, both the SEA and the cross-correlation analysis consistently show that the minimum in SPH
precedes the maximum in solar forcing by a few days, indicating the action of other forcings or
atmospheric effects.

Second, not only the SPH sensitivity to solar forcing is larger for periods of low solar activity,
even the amplitude of the potential solar 27-day signature is larger during solar minimum, which is
currently not understood at all. Interestingly, Gruzdev et al. (2009) find in their HAMMONIA model
studies generally a non-linear atmospheric response to solar forcing with sensitivities increasing
with decreasing forcing. This is in part consistent with our results. However, Gruzdev et al. (2009)
emphasize that the amplitude of the atmospheric response does not increase with decreasing forcing,
which is inconsistent with our results on the SPH response to solar forcing. The apparent increase
in the amplitude of the potential 27-day signature in SPH with decreasing solar activity may also
be an artifact and caused by effects unrelated to solar variations. If this is the case, it is, however,
unexpected that the phase relationship between solar forcing and the potential response in SPH
essentially remains the same, independent of solar activity. This could be a synchronization effect.
In this context it is important to mention that Ebel et al. (1981) performed a cross-spectral analysis
of the solar F10.7 cm flux and planetary wave activity at pressure levels between 10 and 50 hPa.
They found significant correlations between solar variability and the amplitude of planetary waves.

Third, the amplitude of the potential solar 27-day signature in SPH is about a factor of two larger
during winter than during summer. It is well known that due to the winter anomaly the SPH am-
plitudes are increased in winter by larger downward transport of NO from the thermosphere and



subsequent photo-ionisation (e.g., Peters et al., 2017; Garcia et al., 1987)). Garcia et al. (1987) examined the electron density anomalies in the boreal D region in a coupled model with neutral and ion photochemistry, as well as transport by planetary waves. They found that anomalies can be understood in terms of auroral production of nitric oxide in polar night and its subsequent transport and ionization. In particular, their results indicate the importance of horizonal ultra-long planetary wave

transport for many of the observed features. In addition, Hendricks et al. (2015) clearly demonstrated the impact of the 27-day solar cycle on NO production in the Auroral zone in satellite measurements during events of energetic particle precipitation (EPP). The authors found larger amplitudes of the EPP-driven 27-day signature in NO during winter than during summer, which may contribute to the larger amplitudes of the 27-day signatures in SPH reported here. Gruzdev et al. (2009) also discuss

seasonal variations of the atmospheric response to the solar 27-day cycle. For extra-tropical latitudes they report that the sensitivities are for many parameters larger in winter than in summer.

In order to investigate the influence of planetary waves, temperature data taken from the ERA-Interim Reanalysis as well as from model simulations with a nudged version of CMAM were used in the current study. The presented results provide clear evidence that planetary waves are associated

with spectral power in the quasi 27-day period range and lead to corresponding variations in SPH. The different analysis techniques provide complementary approaches to investigate different sources of variability in SPH. While the SEA allows a robust identification of a solar-driven 27-day signature, the regression analysis applied to SPH and CMAM GH allows separating dynamical effects. The presented investigations allowed improving the scientific understanding of several aspects of solar

and dynamical influences on SPH. However, an overall and coherent picture is still missing, as several of the reported effects are difficult to quantify and understand. In addition, a potential impact of solar variability on planetary wave activity is not well understood.

In the context of 27-day variations in SPH it is also relevant that a solar 27-day signature in noctilucent cloud (NLC) altitude was recently discovered (Thurairajah et al., 2017; Köhnke et al.,

2018). The signature has an amplitude of about 100 – 200 m. Köhnke et al. (2018) provide a qualitative explanation for phase relationship of the identified 27-day signature in NLC altitude, NLC occurrence rate and temperature at the polar summer mesopause. The 27-day signature in NLC parameters is likely mainly driven by dynamical effects (see Köhnke et al. (2018)). The main reason is that the phase relationship between the 27-day signatures in temperature and $H_2O$ mixing ratio at

the summer mesopause found by Thomas et al. (2015) is inconsistent with a purely photochemical process, but easily explained by a solar modulation of the upwelling in the polar summer mesosphere.

Further insight into the underlying processes may be gained by dedicated model simulations using a general circulation model, coupled to an ion chemical module capable of modelling all relevant physical (particularly dynamical) and chemical processes.



## 6 Conclusions

We identified for the first time a solar-driven 27-day signature in standard phase height (SPH) measurements. Employing a Monte-Carlo approach, the 27-day solar cycle signature was shown to be highly significant. SPH is anti-correlated to the solar forcing (at the 27-day scale), but the phase height minimum occurs a few days before the solar maximum, indicating that the 27-day solar cycle signature in standard phase heights is not only a consequence of variable photo-ionisation of NO. We argue that non-trivial dynamical effects potentially cause the observed phase lags. The exact mechanisms are, however, currently unknown. It was demonstrated that both the sensitivity of standard phase heights to solar forcing at the 27-day scale, as well as the amplitude of the 27-day signature depend on several parameters, including solar activity, season and the specific prior treatment of the time series. If the entire time series is analyzed, the 27-day signature in standard phase height has an amplitude of about 50 m and a 27-day sensitivity value is obtained, that agrees within combined uncertainties with the sensitivity value of the 11-year solar cycle (i.e., -0.436 ($\pm$ 0.049) km (100 sfu)$^{-1}$). The latter value is in agreement with the study by Peters and Entzian (2015). For the first time the presented regression results provide clear evidence that planetary waves are associated with quasi 27-day periods and lead to corresponding SPH variations in the extratropics during solar minimum in winter. Several findings are unexpected and currently not fully understood. A full understanding of these effects requires dedicated model simulations considering all relevant physical and chemical processes.

*Acknowledgements.* The authors are indebted to the Leibniz-Institute of Atmospheric Physics in Kühlungsborn and the precursor institutions for running the phase height measurements over more than 5 decades. We are grateful to the many technicians who were involved in maintaining the recordings. We thank Frau Wecke for plotting Figure 12 and Jorge Chau for carefully reading the manuscript and for making suggestions for improvements. We wish to thank Deutscher Wetterdienst, ECMWF and the Royal Observatory (SILSO, 2017; SIDC, Version 2) of Belgium and LISIRD of Colorado University (USA) for providing reanalysis data and the solar Lyman-$\alpha$ and F10.7 cm time series, respectively. This work was supported by the University of Greifswald and by the German Ministry of Education and Research (BMBF) through ROMIC (Role Of the Middle atmosphere In Climate) project OHCycle (Grant 01LG1215A).



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
