# Peer review of "Solar 27-day signatures in standard phase height measurements above central Europe"

_Atmospheric Chemistry and Physics, 2018_

## Short Comment (SC1) · 27 Aug 2018

You observed remarkably larger amplitude of the 27-day variation in the phase reflection heights near 80 km at solar minimum than at solar maximum. You do not have explanation for this finding. Let me inform you that the remarkably larger amplitude of the 27-day variation in the lower ionosphere was also found in the radio wave absorption in the lower ionosphere by Pancheva et al. (1991). Main conclusions nfrom this paper: The 27-day fluctuations in the lower ionosphere are of direct solar origin only if the Lyman-$\alpha$ flux exhibits a very well expressed solar rotation variation. The absorption fluctuations are largest in winter near solar activity minimum, in fair coincidence with the maxima of corresponding fluctuations in zonal and particularly meridional winds. This indicates a dynamical forcing (maybe of solar origin).

[Figure]

Reference: D. Pancheva, R. Schminder, J. Lastovicka (1991): 27-day fluctuations in the ionospheric D-region. J. Atmos. Terr. Phys., 53 (11/12), 1145-1150, https://doi.org/10.1016/0021-9169(91)90064-E.

Best regards, Jan Lastovicka
* * *

---

## Referee Comment (RC1) · Anonymous Referee #3 · 30 Aug 2018

This is an interesting paper that is potentially appropriate for ACP. The authors present a thorough uncertainty analysis for their SEA approach and I think their basic results are credible. They also find some interesting deviations from simple expectations- which are somewhat credibly presented; however, while I appreciate their difficulties in trying to explain all this, I confess I got lost in trying to understand much of their hypothesizing. I had difficulties with Figures 11 and 12 to the extent that I do not see where they can state on lines 429-430 that "for the first time . . ... associated with quasi 27 day periods". For that to be valid, I need to see a Fourier spectrum of geopotential height with significant power at that period. In other words, Figure 3 needs to be repeated for GPH (preferably as a function of altitude, as per some of my comments below). And even if they do that- did not they just say that this was first shown by Ebel

et al, 1981. So what justifies the phrase "for the first time"?

Other Major comments

A. Writing/presentation: I recommend breaking up Section 4. It's a jumble of analyses that comes off confusingly. Tellingly, they have to subdivide their Section 3 times (4.2.2) which is hard to follow. They should have a section on "Results" which present their 4 basic results (i.e. SPH basically correlates plus the three puzzles as listed in Section 5). Then Section 4.3 is really (I think) an attempt to find some interpretation- this should be separated.

B. Figure 11 confused me. First, (line 284), there is no "middle panel". Only top and bottom. Second, where is SPH in all this- why can't they correlate the CMAM .01 hPA temperature with SPH? Third, and related, what is the altitude variation of the variability in this band-pass? Or altitude variation of the correlation/regression with SPH? This would relate to whether the forcing is in-situ (i.e. planetary wave mixing at .01 hpa) or due to integrated height changes.

C. The issue of the poorly understood negative lag. First, where do they show this? Which figure has the correlation plotted vs. phase shown a peak at a specific phase? In the absence of this, where am I supposed to find the phase lag? All I see is some words on line 168-169. Does the phase lag change in winter vs. summer? Their arguments in 4.3.2 would seem to be relevant for winter (i.e. requires a mesospheric vortex). Are they saying that the effect is so small in summer that they are ignoring it? That may be OK, but if so, say so more explicitly.

D. Note, there is literature on this question dating back to ozone studies in the 1980s. See for example, Brasseur et al., JGR, 1987, page 903 or Eckman, JGR, 1986, page 6705. Mathematically, from Fourier analysis, if there is damping or negative feedback, it will manifest itself as a negative lag (i.e. response precedes forcing). I confess I do not know if this shows up in wavelets, but it's worth considering.

E. Why are they choosing a phase lag of 12 days for Figure 12? Shouldn't they use the phase lag for which the correlation maximizes? Earlier in the text they say 1-3 days.

F. I looked at the CMAM30 web site they give. There is nitric oxide data. I suggest they use this data to compare with geopotential height, solar changes etc.

Minor comments

1. They need to specify where they got the Lyman alpha time series. Is this a proxy they developed? Is it from satellite data?

2. It would be helpful to provide more context to the standard height technique. I realize this technique is mature, but there are also VLF measurements which, at first glance, are pretty similar in approach. In Peters and Entzian, they mention a reflection height of 500 cm-3. Does the shape of the profile matter? It does for VLF.

3. The VLF technique provides an altitude profile- see any number of papers by N.R. Thomson. Apparently the SPH technique does not? But this makes it hard to interpret height changes. What does a 1 km height change really mean in terms of the electron density? Is this a local increase, or descent of a layer?

4. Lines 322-332. Very confusing. What is "it follows" (line 327)? Sentence needs a verb. Then I don't understand the argument on lines 331. Why should electron density go up if air pressure is higher? Perhaps it would mean more recombination and thus the opposite. And this sounds like a different mechanism than on 324- southward transport.

5. Lines 324-354: There are references worth citing on mesospheric nitric oxide transport and planetary waves, for example: Siskind et al., JGR, 1997, p.3527. Mesospheric transport due to breaking planetary waves is also covered in Sassi et al., JGR, 2002, 4380. More recently, work by Lynn Harvey has discussed the mesospheric polar vortex. She uses CO as a diagnostic. I don't suggest they redo her work, but certainly consider it and cite it, at minimum.

6. Figure 12 needs color bars. None of the labels are readable. The caption should explicitly state what the red/pink and blue colors are. My evaluation of this figure and associated text will likely change once I can actually make out what this is a plot of.

7. Why do they choose 12 days for $\frac{1}{2}$ a solar cycle (lines 315-316)? Should be 13 or 14. (but also consider comment E. above).

―――――――――――――――

---

## Referee Comment (RC2) · Anonymous Referee #2 · 14 Sep 2018

The authors use Kühlungsborn phase height measurements to analyse the response of the D region / upper mesosphere to the 27 day solar cycle. They find a negative correlation as expected, seasonal and solar cycle effects, and, unexpectedly, a stronger effect during solar minimum than during solar maximum. There is a lag of few days between solar variability and (negative) phase height. They also use CMAM geopotential heights to gain some insight into possible atmospheric effects on the phase height variability. The results are new and interesting, the analysis is made with care, and I recommend publication of the paper.

I found section 4.3.2 a bit unsatisfactorily. The authors present regression of phase heights and CMAM geopotential height maps at lag zero. Why was this lag chosen? Is there a lag between CMAM geopotential heights near the reflection point and the

solar flux? The authors propose NO transport for being responsible for the negative correlation of phase heights and geopotential heights. Since NO or winds are not shown, this is only qualitative. However, given this effect is working, phase heights should decrease as long as transport anomalies are southward, and a delay between geopotential heights and phase heights could be compared with this analysis, which could be based on gradient wind anomalies, or simply visual inspection of CMAM maps at different lag.

Minor comments

L 85: I do not see a 27-day peak in Figure 3, but enhanced power also at about 23 days.

L 95: The data analysis is the same than in section 3.1, but the whole time series is used, right?

L 106: Have there been limitations concerning the solar flux maxima, e.g., amplitude, or distance between them?

L 109: replace "N" by "N = 584".

L 156: I do not understand, phase height and LYA have different units and amplitudes, what is the meaning of the ratio?

Fig. 6: Axis labels for LYA and solar flux should be added.

L 160: I see the negative correlation for the summer of 1985, but not for 1986.

L 205: Maybe mention that the correlation between F10.7 and LYA does not hold for solar minimum.

L 266: Insert a period after the URL.

Caption of Fig. 11: The bottom part refers to 0.01 hPa.

L 289: middle panel -> lower part of the upper panel.

[Figure]

L 312: Days -> days

L 502: Add an URL

---

## Referee Comment (RC3) · Anonymous Referee #1 · 20 Sep 2018

This is an interesting and well-designed study of 27-day solar UV-induced variations of "standard phase height" (SPH) measurements in the ionospheric D region. While the observed inverse correlation of SPH with 27-day variations of solar EUV flux and the agreement of derived sensitivities with those obtained on the 11-year time scale are consistent with a purely photochemical mechanism (enhanced photo-ionisation of NO leading to lower effective reflection heights), a number of characteristics of the response are not consistent with such a mechanism. This leads the authors to further analyses identifying "non-trivial dynamical effects" and "clear evidence that planetary waves are associated with quasi 27-day periods". After revisions in response to comments by the other two reviewers and by Jan Lastovicka as well as my comments, publication is recommended in ACP.

[Figure]

(1) One major disagreement with expectations for a purely photochemical origin is the fact (lines 358-360) that "the minimum in SPH precedes the maximum in solar forcing by a few days". This statement agrees with the phase shift listing in Table 1, which shows shifts ranging from -2 to -4 days with no apparent dependence on solar activity level (F10.7 flux threshold). The negative phase lag of a few days can also be discerned in Figure 4, which shows the superposed epoch analysis (SEA) results for all available data (584 epochs). However, this is not so clear from the SEA results for low solar activity periods. Looking at the smooth red curve in the lower panel of Figure 10, the minimum in the SPH anomaly shows no significant negative lag when epochs with F10.7 less than 130 sfu are considered. Please explain.

(2) A second major disagreement is that (lines 361-362) "not only the SPH sensitivity to solar forcing is larger for periods of low solar activity, even the amplitude of the potential solar 27-day signature is larger during solar minimum". The authors go on to note that a model study by Gruzdev et al. (2009) did find increased sensitivities with decreased forcing but did not find an amplitude increase with decreasing forcing. The main evidence for an amplitude increase seems to come from a combination of Figure 10 (showing the SEA results for high and low solar activity epochs) and Table 1, which lists the results as a function of the threshold value of F10.7 used to select the epochs that are analyzed. The lower panel of Figure 10 (F10.7 < 130 sfu) does indeed show a larger amplitude of the 27-day SPH variation than the upper panel (F10.7 > 130 sfu). However, this difference is not so clear from Table 1. The top entry corresponds to all data (F10.7 threshold of at least 60 units, 584 epochs) and gives an amplitude of 47.8 meters. The eighth entry corresponds to all data when F10.7 was at least 130 sfu (303 epochs) and gives an amplitude of only 31.5 meters, which is roughly consistent with the upper panel of Figure 10. However, the next entry in the table corresponds to data when F10.7 was at least 140 sfu (267 epochs) and gives an amplitude of 40.5 meters, considerably larger than that when 130 sfu is used as the threshold and maybe not significantly less than the 47.8 meter value obtained when all data are considered. So, I would ask the authors to re-examine their conclusion that the amplitude is larger

under solar minimum conditions.

(3) The listings in Table 2 (summarizing the results for different times of the year) are hard to understand. The mean 27-day sensitivities are larger for the summer (May to Sept.) and winter (Oct. to Feb.) seasons (-.45 and -.49, respectively) than they are for the whole year (-.37). Is this because the sensitivities in March and April are much lower than for the rest of the year? Please explain. Also, the amplitude of the SPH change is about a factor of 2 larger in the winter season than in the summer season even though the mean sensitivities in the two seasons are about the same. The discussion in section 5 (lines 375-389) gives some explanation for why the amplitude should be larger in winter, involving larger downward transport of NO from the thermosphere, increased 27-day NO production in the Auroral zone, etc. But shouldn't the sensitivities be larger in winter also?

(4) One standard deviation error limits are shown in the figures (e.g., Figure 4). If I remember correctly, the probability that the true value lies within these limits is only around 68%. Two standard deviation limits corresponds to about 95% confidence. I would not insist on changing the plotted limits to two standard deviations in the figures but at least it should be mentioned in the text that two standard deviations would provide a better indicator of whether a real signal is being detected. For example, in the upper panel of Figure 10, the peaks in the mean amplitude curve are not significantly different from the minimum near zero lag if two standard deviation limits are used.

Minor Points:

(6) This is probably just personal preference but I would use "inversely correlated" rather than anti-correlated, which could be misinterpreted to mean no correlation.

(7) It is necessary to read the details of section 2 to understand what standard phase height means. Since this manuscript is apparently submitted to a special issue on layered phenomena in the mesopause region, it may be obvious to most readers that this term refers to an effect in the upper atmosphere. However, to save many casual

readers the trouble of reading section 2 to understand what the manuscript is about, it would be better to indicate in the abstract (preferred) or introduction what is being investigated. For example, in the abstract, you could change line 2 to read: "standard phase height measurements in the ionospheric D region carried out in central Europe."

(8) Line 86: ... such as the atmospheric processes ...

(9) Line 130: ... by checking how many of the ...

(10) Lines 329-330. Please re-write this sentence.

---

## Author Comment (AC1) · 21 Dec 2018

**Reply to comments by reviewer #1**

**Note: our replies are boldfaced**

Reviewer comment: This is an interesting and well-designed study of 27-day solar UV-induced variations of "standard phase height" (SPH) measurements in the ionospheric D region. While the observed inverse correlation of SPH with 27-day variations of solar EUV flux and the agreement of derived sensitivities with those obtained on the 11-year time scale are consistent with a purely photochemical mechanism (enhanced photo-ionisation of NO leading to lower effective reflection heights), a number of characteristics of the response are not consistent with such a mechanism. This leads the

authors to further analyses identifying "non-trivial dynamical effects" and "clear evidence that planetary waves are associated with quasi 27-day periods". After revisions in response to comments by the other two reviewers and by Jan Lastovicka as well as my comments, publication is recommended in ACP.

***Reply: We thank the reviewer for his/her encouraging comments.***

(1) One major disagreement with expectations for a purely photochemical origin is the fact (lines 358-360) that "the minimum in SPH precedes the maximum in solar forcing by a few days". This statement agrees with the phase shift listing in Table 1, which shows shifts ranging from -2 to -4 days with no apparent dependence on solar activity level (F10.7 flux threshold). The negative phase lag of a few days can also be discerned in Figure 4, which shows the superposed epoch analysis (SEA) results for all available data (584 epochs). However, this is not so clear from the SEA results for low solar activity periods. Looking at the smooth red curve in the lower panel of Figure 10, the minimum in the SPH anomaly shows no significant negative lag when epochs with F10.7 less than 130 sfu are considered. Please explain.

***Reply: The reviewer is right and the phase lag for solar minimum is smaller, and there may be no phase lag at all. We currently don't have a plausible explanation for this behavior and we do not claim to have one. We can't either say with certainty, why the minimum in SPH occurs a few days before the maximum in solar activity, if the entire time series is analyzed. We believe that non-trivial dynamical effects are the cause of these phase lags. We added a brief statement to section 4.2 of the manuscript. We would like to point out that the phase lags listed in Table 1 are not inconsistent with the lower panel of Figure 10. Table 1 shows the analysis results for epochs with solar activity exceeding a certain F10.7 cm flux threshold, i.e. the case that solar activity is lower than a certain value is not shown in the Table.***

(2) A second major disagreement is that (lines 361-362) "not only the SPH sensitivity

to solar forcing is larger for periods of low solar activity, even the amplitude of the potential solar 27-day signature is larger during solar minimum". The authors go on to note that a model study by Gruzdev et al. (2009) did find increased sensitivities with decreased forcing but did not find an amplitude increase with decreasing forcing. The main evidence for an amplitude increase seems to come from a combination of Figure 10 (showing the SEA results for high and low solar activity epochs) and Table 1, which lists the results as a function of the threshold value of F10.7 used to select the epochs that are analyzed. The lower panel of Figure 10 (F10.7<130 sfu) does indeed show a larger amplitude of the 27-day SPH variation than the upper panel (F10.7>130 sfu). However, this difference is not so clear from Table 1. The top entry corresponds to all data (F10.7 threshold of at least 60 units, 584 epochs) and gives an amplitude of 47.8 meters. The eighth entry corresponds to all data when F10.7 was at least 130 sfu (303 epochs) and gives an amplitude of only 31.5 meters, which is roughly consistent with the upper panel of Figure 10. However, the next entry in the table corresponds to data when F10.7 was at least 140 sfu (267 epochs) and gives an amplitude of 40.5 meters, considerably larger than that when 130 sfu is used as the threshold and maybe not significantly less than the 47.8 meter value obtained when all data are considered. So, I would ask the authors to re-examine their conclusion that the amplitude is larger under solar minimum conditions.

***Reply: Our manuscript is indeed a bit misleading in this context, we apologize. The lower panel of Fig. 10 cannot be directly compared to Table 1. Table 1 lists the analysis results for solar activity exceeding a certain F10.7 cm flux threshold, while the bottom panel of Fig. 10 shows the SEA results for epochs with F10.7 cm values less than 130 sfu. In other words, for the two analyses, different sets of epochs are used. We now added some text to section 4.2 to address this point explicitly.***

(3) The listings in Table 2 (summarizing the results for different times of the year) are hard to understand. The mean 27-day sensitivities are larger for the summer (May to

Sept.) and winter (Oct. to Feb.) seasons (-.45 and -.49, respectively) than they are for the whole year (-.37). Is this because the sensitivities in March and April are much lower than for the rest of the year? Please explain. Also, the amplitude of the SPH change is about a factor of 2 larger in the winter season than in the summer season even though the mean sensitivities in the two seasons are about the same. The discussion in section 5 (lines 375-389) gives some explanation for why the amplitude should be larger in winter, involving larger downward transport of NO from the thermosphere, increased 27-day NO production in the Auroral zone, etc. But shouldn't the sensitivities be larger in winter also?

*Reply: We cannot fully explain some of the obtained results, as already mentioned several times in the manuscript. This is certainly unsatisfactory, but we hope that some of these aspects will be better understood in the future, when the understanding of the relative contributions of dynamics (potentially solar driven) and solar variability has been improved. We now also cite and briefly discuss the study by Pancheva et al. (1991) reporting on quasi 27-day variations in radio wave absorption measurements. The reported results are quite consistent with many of our puzzling results, e.g. larger amplitudes during winter and solar minimum.*

*In addition, we would like to point out that the sensitivity values for summer, winter and all-year agree within a 2-sigma range. It is difficult to state, whether the sensitivities are much lower in March and April, because an analysis of these months would be based on a small number of epochs.*

*The relatively large fraction of random ensemble members with amplitudes larger than the actual analysis for summer (11.2%) may be a reason for the unexpected behavior of amplitudes vs. sensitivities between summer and winter. This is now explicitly stated in the manuscript.*

(4) One standard deviation error limits are shown in the figures (e.g., Figure 4). If I

remember correctly, the probability that the true value lies within these limits is only around 68

*Reply: The thin lines in Figure 4 (both for standard phase height and the F10.7 cm flux) do not correspond to one standard deviation, but are the standard errors of the mean, i.e. the standard deviation divided by the number of epochs. The standard deviations themselves are much larger than the values shown. The top panel of Figure 2 also shows the standard deviations of the SPH anomaly, which is on the order of 600 m. The standard errors of the mean shown, e.g., in Figure 4 are determined the following way: For each day relative to local solar maximum the corresponding SPH anomaly values from the individual epochs are averaged and their standard deviation is determined. The standard error of the mean is the standard deviation divided by the square root of the number of epochs. This is standard procedure and this is the correct estimate of the uncertainly of a mean value.*

**Note: point (5) was missing in the review**

Minor Points:

(6) This is probably just personal preference but I would use "inversely correlated" rather than anti-correlated, which could be misinterpreted to mean no correlation.

*Reply: OK, changed throughout the manuscript*

(7) It is necessary to read the details of section 2 to understand what standard phase height means. Since this manuscript is apparently submitted to a special issue on layered phenomena in the mesopause region, it may be obvious to most readers that this term refers to an effect in the upper atmosphere. However, to save many casual readers the trouble of reading section 2 to understand what the manuscript is about, it would be better to indicate in the abstract (preferred) or introduction what is being investigated. For example, in the abstract, you could change line 2 to read: "standard

phase height measurements in the ionospheric D region carried out in central Europe."

*Reply: OK, changed. We also added some additional pieces of information to this sentence that hopefully make the concept of phase heights easier to understand.*

(8) Line 86: ... such as the atmospheric processes ...

*Reply: changed (assuming the comment refers to line 88 in the ACPD version)*

(9) Line 130: ... by checking how many of the ...

*Reply: the suggested change does not fit to the rest of the sentence. We changed the sentence to*

*"by checking for how many of the 1000 random cases the amplitude of the fitted sinusoidal function equals or exceeds the amplitude of the sinusoidal fit to the actual epoch-averaged standard phase height anomaly."*

(10) Lines 329-330. Please re-write this sentence.

*Reply: we revised the entire paragraph, also following the suggestions by reviewer #3 and hope this is now easier to understand. The new paragraph is:*

*"In the upper mesosphere, the negative regression pattern between the GH anomaly and the 24 – 31 day-band-pass filtered SPH time series over central Europe in about 80 km altitude for lag zero may be explained by horizontal planetary wave transport. An increase (decrease) of NO density is caused by southward (northward) transport of NO by ultra-long waves for an observed mean positive latitudinal NO gradient in a region between a high and low (low and high) pressure system. Vertical transport of NO by lifting or subsidence is assumed to be weak, diffusion too. The consequence is an increase (decrease) of the free electron number density due to photo-ionization as discussed by von Cossart and Entzian (1976) with a lower (higher) SPH. That implies that SPH shows a negative*

*(positive) regression with the GH anomaly on the easterly (westerly) side of the high-pressure center."*

---

## Author Comment (AC2) · 21 Dec 2018

**Reply to comment by Jan Lastovicka**

Reviewer comment: You observed remarkably larger amplitude of the 27-day variation in the phase reflection heights near 80 km at solar minimum than at solar maximum. You do not have explanation for this finding. Let me inform you that the remarkably larger amplitude of the 27-day variation in the lower ionosphere was also found in the radio wave absorption in the lower ionosphere by Pancheva et al. (1991). Main conclusions nfrom this paper: The 27-day fluctuations in the lower ionosphere are of direct solar origin only if the Lyman-alpha flux exhibits a very well expressed solar rotation variation. The absorption fluctuations are largest in winter near solar activity

minimum, in fair coincidence with the maxima of corresponding fluctuations in zonal and particularly meridional winds. This indicates a dynamical forcing (maybe of solar origin).

Reference: D. Pancheva, R. Schminder, J. Lastovicka (1991): 27-day fluctuations in the ionospheric D-region. J. Atmos. Terr. Phys., 53 (11/12), 1145-1150, https://doi.org/10.1016/0021-9169(91)90064-E.

Best regards, Jan Lastovicka

*Reply: Many thanks for this information. We now cite the paper by Pancheva et al. (1991) and added the following statements to the discussion section:*

*"In this context it is also important to mention that Pancheva et al. (1991) investigated quasi 27-day fluctuations in ground-based measurements of radio wave absorption in the lower ionosphere. They found indications for several aspects that are in good qualitative agreement with the results presented here. The reported absorption fluctuations are largest during winter near solar minimum, suggesting a dynamical forcing, which may be of solar origin, as the authors suggest."*

---

## Author Comment (AC3) · 21 Dec 2018

**Reply to comments by reviewer #2**

**Note: our replies are boldfaced**

Reviewer comment: The authors use Kühlungsborn phase height measurements to analyse the response of the D region / upper mesosphere to the 27 day solar cycle. They find a negative correlation as expected, seasonal and solar cycle effects, and, un-expectedly, a stronger effect during solar minimum than during solar maximum. There is a lag of few days between solar variability and (negative) phase height. They also use CMAM geopotential heights to gain some insight into possible atmospheric effects on the phase height variability. The results are new and interesting, the analysis is

made with care, and I recommend publication of the paper.

*Reply: We thank the reviewer for this encouraging comment!*

I found section 4.3.2 a bit unsatisfactorily. The authors present regression of phase heights and CMAM geopotential height maps at lag zero. Why was this lag chosen? Is there a lag between CMAM geopotential heights near the reflection point and the solar flux?

*Reply: This is a misunderstanding because the lags belong to different research objectives.*

*On the one hand, there is a negative lag of a few days between (negative) SPH and Lyman-alpha (or F10.7) seen in SEA and in a correlation analysis, but this lag difference is only a hint, that the SPH minima appear before solar maximum, saying that SPH are possibly followed by a solar signal.*

*On the other hand, we used a regression analysis in order to show that the variability of the local SPH time series is statistically linked to the GH anomaly field of the boreal extra-tropics. The regression coefficient field is shown for two lags of regression (-12 and 0 days, Figure 12) in order to demonstrate the oscillation behavior in the NH at an upper mesospheric level (0.01 hPa about 80 km altitude) and in the stratopause region (1 hPa about 48 km). That means we demonstrated that the large-scale regression patterns are changing their sign during a period of about half of the "27 solar period". We also performed the regression analysis for a lag of -15 days, which essentially resulted in the same results as for a lag of -12 days.*

The authors propose NO transport for being responsible for the negative correlation of phase heights and geopotential heights. Since NO or winds are not shown, this is only qualitative.

*Reply: This is correct, but it is a future task which could be examined in model*
*study.*

However, given this effect is working, phase heights should decrease as long as transport anomalies are southward, and a delay between geopotential heights and phase heights could be compared with this analysis, which could be based on gradient wind anomalies, or simply visual inspection of CMAM maps at different lag.

*Reply: The transport time as well as the recombination times are fast so that in our opinion such a visual inspection of CMAM maps is not appropriate. A future sensitivity investigation with a middle atmosphere model including ion chemistry could be used to examine this issue.*

Minor comments

L 85: I do not see a 27-day peak in Figure 3, but enhanced power also at about 23 days.

*Reply: agreed, we changed the statement to*

*"Strong signatures at periods between 55 days and 22 days are identified"*

L 95: The data analysis is the same than in section 3.1, but the whole time series is used, right?

*Reply: correct, the whole time series are now used. We now mention this explicitly.*

L 106: Have there been limitations concerning the solar flux maxima, e.g., amplitude, or distance between them?

*Reply: Good question! No assumptions were made in terms of the amplitude of the solar proxy maxima, but implicitly a constraint has been applied regarding the temporal distance between the maxima: We checked for every day of the time series, whether the corresponding daily value is larger equal than the values in the period from -13 to +13 days around the corresponding day. This way,*

*multiple or minor maxima that are only a few days apart will not be identified. The automated search was applied to the F10.7 cm time series smoothed with a 5-day box-car function, as already mentioned in the manuscript. The identification of the maxima was checked visually and the approach was found to work well. This procedure is now briefly explained in the manuscript.*

L 109: replace "N" by "N = 584".

*Reply: Changed.*

L 156: I do not understand, phase height and LYA have different units and amplitudes, what is the meaning of the ratio?

*Reply: The reviewer is absolutely correct, giving a dimensionless ratio doesn't make sense here. We replaced the statement by:*

*"The SPH amplitudes are typically larger during winter compared to summer and for some solar cycles also appear to be larger during solar minimum than for solar maximum."*

Fig. 6: Axis labels for LYA and solar flux should be added.

*Reply: The figure was revised and the additional figure labels were added.*

L 160: I see the negative correlation for the summer of 1985, but not for 1986.

*Reply: The reviewer is right, in summer 1986 no clear negative correlation is present. We changed the statement to read:*

*"During May-July 1985 the two band-pass-filtered time series are out-of-phase (less obvious during May-July 1986) as expected from photo-ionization by Lyman-alpha, and the larger phase difference change during winter time may be due to atmospheric processes."*

L 205: Maybe mention that the correlation between F10.7 and LYA does not hold for

solar minimum.

***Reply: Good idea, we now mention this and cite Barth et al., Comparison of F10.7 cm radio flux with SME solar Lyman alpha flux, GRL, 17(5), 1990.***

L 266: Insert a period after the URL.

***Reply: Done.***

Caption of Fig. 11: The bottom part refers to 0.01 hPa.

***Reply: the bottom part refers to a 1 hPa. Thanks for pointing out that this was not mentioned. We added this piece of information to the Figure caption.***

L 289: middle panel -> lower part of the upper panel.

***Reply: Sentence adjusted, thanks!***

L 312: Days -> days

***Reply: done.***

L 502: Add an URL

***Reply: We added an URL for this paper, but we're not sure, whether it is consistent with the ACP guidelines. It may be removed again.***

––––––––––––––––––––––––––

---

## Author Comment (AC4) · 21 Dec 2018

**Reply to comments by reviewer #3**

*Note: our responses are indented and bold-faced*

Reviewer comment: This is an interesting paper that is potentially appropriate for ACP. The authors present a thorough uncertainty analysis for their SEA approach and I think their basic results are credible. They also find some interesting deviations from simple expectations which are somewhat credibly presented; however, while I appreciate their difficulties in trying to explain all this, I confess I got lost in trying to understand much of their hypothesizing.

> *Reply: Thanks for pointing this out. We tried to improve our explanations and hope that our reasoning is now easier to follow (please see specific responses below).*

I had difficulties with Figures 11 and 12 to the extent that I do not see where they can state on lines 429-430 that "for the first time : : :.. associated with quasi 27 day periods".

> *Reply: We changed the text in the following way in order to be more specific:*

> *"To our knowledge, we demonstrated for the first time the possible link between band-pass filtered (24-30 days) SPH time series and large-scale geopotential height fields in the extra-tropical boreal upper stratosphere and mesosphere during solar minimum. We identified a planetary wave 1/wave 2 structure in the regression-coefficient distribution at the stratopause and the upper mesosphere showing an oscillation pattern with a period of about 27 days."*

For that to be valid, I need to see a Fourier spectrum of geopotential height with significant power at that period. In other words, Figure 3 needs to be repeated for GPH (preferably as a function of altitude, as per some of my comments below).

> *Reply: We calculated the CMAM-GH spectrum over the Eifel mountains at four relevant altitudes, namely: 10, 1, 0.1, 0.01 hPa, shown in Figure-X3 below. We found significant power in the range of periods between 24 and 30 days, which was expected from Figure 11, bottom panel. This is also in good agreement with Schanz et al. (2016).*

> *We add a sentence to explain that we calculated the spectrum (not shown) and saying the we found significant power in the range of periods between 24 and 30 days, which was expected from Figure 11, bottom panel.*

> *Schanz, A., K. Hocke, and N. Kämpfer, On forced and free atmospheric oscillations near the 27-day periodicity, Earth, Planets and Space, 68:97, https://doi.org/10.1186/s40623-016-0460-yS, 2016.*

[Figure]

*Figure-X3. Power spectrum of CMAM GH over Eifel mountains.*

And even if they do that- did not they just say that this was first shown by Ebel et al, 1981.

> *Reply: Ebel et al. (1981) were among the first who showed that the solar signal influences the planetary wave signal in the stratosphere, but didn't study the mesosphere. We now specified our statement:*
>
> *"In this context it is important to mention that Ebel et al. (1981) performed a cross-spectral analysis of the solar F10.7 cm flux and planetary wave activity in the stratosphere at pressure levels between 10 and 50 hPa."*
>
> *Note the abstract of Ebel et al. (1981): "The effects of solar activity on the geopotential height and temperature fields of the 50-, 30- and 10-mbar surface, resolved into zonal harmonic components, were investigated. This was done by means of cross-spectral analysis between the 10.7-cm radiation of the sun and planetary waves up to zonal wave number 3. Frequent significant responses of various harmonic components in a broad range of oscillation frequencies give evidence that solar activity plays a significant role for the dynamics of the middle and lower stratosphere. Oscillations of the amplitudes of the zonal harmonics that are coherent with solar activity fluctuations were extracted from the spectra and recomposed into coherent (planetary) waves. Three waves with periods of 25 days (near to the sun's rotation period), 13.6 days (first harmonic of solar rotation), and 15.1 days (corresponding to the well known 15- to 16-day wave in the atmosphere) are examined in detail. They show the properties of free planetary modes (13.6 and 15.1 days) and possibly of internal waves (25 days) at higher latitudes. Vacillation cycles of the mean atmospheric state (including stationary waves) seem to be important for the generation of the studied wave phenomena."*

So what justifies the phrase "for the first time"?

> *Reply: These lines were re-written for clarity. See also comments above.*

Other Major comments

A. Writing/presentation: I recommend breaking up Section 4. It's a jumble of analyses that comes off confusingly. Tellingly, they have to subdivide their Section 3 times (4.2.2) which is hard to follow. They should have a section on "Results" which present their 4 basic results (i.e. SPH basically correlates plus the three puzzles as listed in Section 5). Then Section 4.3 is really (I think) an attempt to find some interpretation- this should be separated.

>*Reply: We agree there are probably too many subsections in section. We tried to follow the reviewer's suggestion by*

>>*a) Removing the subsections of section 4.2*
>>*b) Moving section 4.3 to a new section 5.*

B. Figure 11 confused me. First, (line 284), there is no "middle panel". Only top and bottom.

>**Reply:** *We apologize, there is indeed no middle panel, this is now corrected.*

Second, where is SPH in all this- why can't they correlate the CMAM .01 hPA temperature with SPH?

>*Reply: The SPH series is shown in Figure 1 without any filtering; in Figure 2 it is shown after applying 41-day running mean and in Figure 6 and in Figure 7 for the period 1985-87.*

>*The correlation between SPH and CMAM-0.01-hPa-T is -0.105, and the correlation between band pass filtered SPH and CMAM-0.01-hPa-T is -0.013, both are very weak.*

Third, and related, what is the altitude variation of the variability in this band-pass?

>**Reply:** *The altitude variation in this GH band pass at 0.01 hPa is about ±0.5 km, and at 1 hPa about ± 1 km.*

Or altitude variation of the correlation/regression with SPH? This would relate to whether the forcing is in-situ (i.e. planetary wave mixing at .01 hpa) or due to integrated height changes.

>*Reply: The values of regression between band pass filtered SPH and unfiltered GH are shown in Figure 12 with larger contour labels.*

C. The issue of the poorly understood negative lag. First, where do they show this? Which figure has the correlation plotted vs. phase shown a peak at a specific phase? In the absence of this, where am I supposed to find the phase lag? All I see is some words on line 168-169. Does the phase lag change in winter vs. summer? Their arguments in 4.3.2 would seem to be relevant for winter (i.e. requires a mesospheric vortex). Are they saying that the effect is so small in summer that they are ignoring it? That may be OK, but if so, say so more explicitly.

>*Reply: The negative lag (or shift) is shown and discussed several times throughout the manuscript, e.g. in Figure 4, Table 1 (section 4.2.2). We did not show the results of the time-lagged cross-correlation, because they are consistent with the SEA results and provide no additional information. We believe that the confusion is in part due to the fact that we use*

*the term "shift" in parts of the manuscript and "lag" in other parts. We now made this more consistent and only use the term "lag" throughout the paper.*

*Regarding seasonal differences in lag: this is discussed in section 4.2.3 and Table 3 explicitly lists the lags for summer and winter. The surprising aspect is that the lag is the same for both seasons.*

*We hope that by using one term (lag), these connections are now less confusing.*

D. Note, there is literature on this question dating back to ozone studies in the 1980s. See for example, Brasseur et al., JGR, 1987, page 903 or Eckman, JGR, 1986, page 6705. Mathematically, from Fourier analysis, if there is damping or negative feedback, it will manifest itself as a negative lag (i.e. response precedes forcing). I confess I do not know if this shows up in wavelets, but it's worth considering.

*Reply: Thank you for pointing this out. It is of course difficult to state, whether the temperature feedback on the solar 27-day signature in O3 is of direct relevance for the phase lag in SPH identified in this study. But it is a very interesting example showing that negative phase lags, i.e. response appears to precede the forcing, can occur in the atmosphere. We added this aspect to the discussion and cited the companion papers by Keating et al. (1987) and Brasseur et al. (1987).*

*We do not fully understand the reviewer's comment on wavelets.*

E. Why are they choosing a phase lag of 12 days for Figure 12? Shouldn't they use the phase lag for which the correlation maximizes? Earlier in the text they say 1-3 days.

*Reply: We apologize, a lag of 12 day is a mistake. It should be a lag of -12 days.*

*This is a misunderstanding because the lags belong to different research objectives.*

*On one hand, there is a negative lag of a few days between (negative) SPH and Lyman-alpha (or F10.7). This was shown in the SEA and in a correlation analysis. However, this lag is only a hint, that the SPH minima appear before solar maximum. This reveals that SPH minimum is possibly followed by a maximum of solar signal of ionization.*

*On the other hand, we used a regression analysis in order to show that the variability of local band pass filtered SPH series is statistically linked with the unfiltered GH anomaly field of the boreal extra-tropics. The regression coefficient is shown for two lags of regression (-12 and 0 days, Figure 12) in order to demonstrate the oscillation behavior in the NH at a mesospheric layer (0.01hPa, about 80 km altitude) and in the stratopause (1 hPa, about 48 km). That means we demonstrate that the large-scale regression patterns are changing their sign during a period of about half of the "27 solar period" as expected. We also performed the regression analysis for a lag of -15 days, which essentially resulted in the same results as for a lag of -12 days.*

F. I looked at the CMAM30 web site they give. There is nitric oxide data. I suggest they use this data to compare with geopotential height, solar changes etc.

*Reply: We examined CMAM30 NO values and compared them to Odin/SMR NO measurements already before. We found that the CMAM NO values, e.g., for mid-latitudes in the mesosphere are only about one third of the Odin measurements, and they show a weaker latitudinal NO gradient than the Odin/SMR data. In the light of the Odin/SMR NO values – which compare well with independent observations – the CMAM NO profiles do not appear credible. This implies that an assessment of 27-day oscillations is CMAM NO data is problematic. Due to the applied nudging technique up to 1 hPa we expect that the meteorological fields in the mesosphere are well represented.*

[Figure]

*Figure XA: VMR of NO CMAM mean January 1979-2010*

**We added a sentence to section 4.3 in order to explain why we do not use CMAM NO data:**

**"The CMAM data set used here also includes NO profiles. Unfortunately, the CMAM NO distribution in the mesosphere does not compare well to satellite observations with the Odin/SMR instrument (Kiviranta et al. 2018). This implies that an assessment of 27-day oscillations is CMAM NO data is problematic"**

Minor comments

1. They need to specify where they got the Lyman alpha time series. Is this a proxy they developed? Is it from satellite data?

> *Reply: The Lyman-alpha and F10.7 cm data sets were obtained from the LASTP Interactive Solar Irradiance Datacenter (LISIRD), as already mentioned in the acknowledgements of the manuscript. This Lyman-alpha data set is a composite of different observational Lyman-alpha data sets. The time period between 1947 and the start of the observational data set is covered by a model based on F10.7 data. Following the reviewer's suggestion we now mention the data source in section 3, when the datasets are discussed the first time. Thanks for pointing this out.*

2. It would be helpful to provide more context to the standard height technique. I realize this technique is mature, but there are also VLF measurements which, at first glance, are pretty similar in approach. In Peters and Entzian, they mention a reflection height of 500 cm-3. Does the shape of the profile matter? It does for VLF.

> **Reply:** *The shape of the profile does not matter for SPH measurements. We assume that the ionosphere is an ideal reflecting mirror and that the reflecting electron density is constant during the day (We added a brief statement about this to section 2.). This is justified if the reflection point is far enough from the D-region maximum and does not change its gradient. Like all other phase measurements in optics also this method is ambiguous. By comparison of different long radio propagation paths (up to 7 paths) we could make it unique coming to an electron density of about 500 el/cm³.*

> *The word "indirect" means that we do not measure phases but only field strength amplitudes, i.e. the sum of ground and sky wave: a field strength maximum means a phase difference of zero or 2nπ, a minimum means a phase difference of 2n+1π.*

3. The VLF technique provides an altitude profile- see any number of papers by N.R. Thomson. Apparently the SPH technique does not? But this makes it hard to interpret height changes. What does a 1 km height change really mean in terms of the electron density? Is this a local increase, or descent of a layer?

> *Reply: The SPH method uses a mean daily reflection layer for a long radio wave signal which refers to a layer of constant electron density at about 80 km altitude. A -1 km height changes means that the reflecting layer of constant electron density over the Eifel mountains changes its height by 1 km.*

> *We added the following statement to section 2 : ".. at the reflection point by a simple geometric-optical method assuming the ionosphere to be an ideal reflecting mirror. The distance .."*

4. Lines 322-332. Very confusing. What is "it follows" (line 327)? Sentence needs a verb. Then I don't understand the argument on lines 331. Why should electron density go up if air pressure is higher? Perhaps it would mean more recombination and thus the opposite. And this sounds like a different mechanism than on 324- southward transport.

>*Reply:  We tried to improve this and the previous statements and hope this is now easier to understand. The new sentences are:*

>*"In the upper mesosphere, the negative regression pattern between the GH anomaly and the 24 – 31 day-band-pass filtered SPH time series over central Europe in about 80 km altitude for lag zero may be explained by horizontal planetary wave transport. An increase (decrease) of NO density is caused by southward (northward) transport of NO by ultra-long waves for an observed mean positive latitudinal NO gradient in a region between a high and low (low and high) pressure system. Vertical transport of NO by lifting or subsidence is assumed to be weak, diffusion too. The consequence is an increase (decrease) of the free electron number density due to photo-ionization as discussed by von Cossart and Entzian (1976) with a lower (higher) SPH. That implies that SPH shows a negative (positive) regression with the GH anomaly on the easterly (westerly) side of the high-pressure center."*

5. Lines 324-354: There are references worth citing on mesospheric nitric oxide transport and planetary waves, for example: Siskind et al., JGR, 1997, p.3527. Mesospheric transport due to breaking planetary waves is also covered in Sassi et al., JGR, 2002, 4380. More recently, work by Lynn Harvey has discussed the mesospheric polar vortex. She uses CO as a diagnostic. I don't suggest they redo her work, but certainly consider it and cite it, at minimum.

>*Reply: Thanks, this is an interesting paper dealing with a two-dimensional NO chemistry model (Siskind et al. 1997), and discussed the planetary wave transport versus diffusion. We take it into account and cite it, also the empirical NO model of Kiviranta et al. (2018). The new paper of Lynn Harvey et al. (2018) is also cited.*

>*We changed lines 322-345. (i) first part is now briefly discussed again in the discussion section section, the second part is changed using right lag = -12 days.*

>*"The positive correlation pattern for a lag of -12 days (Figure 12, panel a) follows from the quasi-periodic 27-day oscillation behavior of the ultra-long wave structure. Furthermore, the negative regression for lag of -12 days and the positive regression for lag zero over eastern Europe reveal the cyclic evolution of the ultra-long planetary waves. In the stratopause layer the regression pattern is positively (negatively) correlated to the 24 – 31 day-band-pass filtered SPH time series over the polar region for lag zero (-12 days) – see panels c) and d) of Figure 12 – indicating a polar vortex weakening (strengthening). The vortex weakening is linked with an intrusion of subtropical air into the polar region over the North Atlantic, as known from some major stratospheric warming events in wintertime (e.g., Peters et al., 2014, Harvey et al., 2018). A dominant wave 1 pattern occurs with a strong wave 2. In general, the results reveal an atmospheric influence especially of ultra-long planetary waves on the 24–31day-band-pass filtered SPH time series during wintertime and solar minimum."*

>*Discussion section insert paragraph:*

*"In the upper mesosphere, the negative regression pattern between GH anomaly and the 24 – 31 day-band-pass filtered SPH time series over central Europe in about 80 km altitude for lag zero may be explained by an increase of NO density caused by southward transport of NO by ultra-long waves in an observed mean positive latitudinal NO gradient, in a region between high and low pressure respectively."*

6. Figure 12 needs color bars. None of the labels are readable. The caption should explicitly state what the red/pink and blue colors are. My evaluation of this figure and associated text will likely change once I can actually make out what this is a plot of.

*Reply: Thanks, we changed the size of contour labels and thickness. We added a more detailed explanation to the figure caption: "The contour interval is 100 m/km, reddish (bluish) areas show positive (negative) regression coefficients."*

7. Why do they choose 12 days for a solar cycle (lines 315-316)? Should be 13 or 14. (but also consider comment E. above).

*Reply: We used a regression analysis in order to show that the variability of local SPH series (24-30 day band pass filtered) is statistically linked with the unfiltered GH anomaly field of the boreal extra-tropics. The distribution of the regression coefficient is shown for two lags of regression (-12 and 0 days, Figure 12) in order to demonstrate the oscillation behavior in the NH at a mesospheric layer (0.01 hPa about 80 km altitude) and in the stratopause (1 hPa about 48 km). That means we demonstrated that the large-scale regression patterns are changing their sign during a period of about half of the "24-30 days band of solar variability. The regression figures were produced for varying lags in steps of 3 days, e.g., 9, 12 and 15. in order to cover a large range of possible lags. The results for a lag of -15 days are very similar compared to a lag of -12 days. The Figure-X12 shows the regression for a lag of -15 days which looks similar in comparison to that of lag -12. We added a statement to point out that the results are quite similar if a lag of -15 days is assumed.*

[Figure]

*Fig.-X12. Regression (in m/km) between the SPH (24 – 31 day-filtered) time series and the unfiltered tine series of geopotential height of CMAM at 0.01 hPa for time lags of -15 days for the period 10/1985 to 4/1986. Contour interval is 100 m/km, reddish (bluish) area showing positive (negative) regression coefficients.*